# PipelineRL: Faster On-policy Reinforcement Learning for Long Sequence Generation

## Abstract

Reinforcement Learning (RL) is increasingly utilized to enhance the reasoning capabilities of Large Language Models (LLMs). However, effectively scaling these RL methods presents significant challenges, primarily due to the difficulty in maintaining high AI accelerator utilization without generating stale, off-policy data that harms common RL algorithms. This paper introduces PipelineRL, an approach designed to achieve a superior trade-off between hardware efficiency and data on-policyness for LLM training. PipelineRL employs concurrent asynchronous data generation and model training, distinguished by the novel *in-flight weight updates*. This mechanism allows the LLM generation engine to receive updated model weights with minimal interruption during the generation of token sequences, thereby maximizing both the accelerator utilization and the freshness of training data. Experiments conducted on long-form reasoning tasks using 32 H100 GPUs demonstrate that PipelineRL achieves approximately $\sim 2x$ faster learning compared to conventional RL baselines while maintaining highly on-policy training data. A scalable and modular open-source implementation of PipelineRL is also released as a key contribution.

## 1 Introduction

Reinforcement Learning (RL) has recently become a popular tool to enhance the reasoning and agentic capabilities of Large Language Models (LLMs) [Guo et al., 2025, Wei et al., 2025]. While RL expands the range of training signals one can use to enhance LLMs, this advanced learning paradigm comes with extra challenges, including being particularly hard to effectively scale to more compute. The scaling difficulty arises from the fact that AI accelerators (like GPUs and TPUs) deliver high throughput only when generating sequences at a large batch size. Hence, naively adding more accelerators to an on-policy RL setup brings increasingly diminishing learning speed improvements because the per-accelerator throughput decreases, while the overall generation latency reaches a plateau. The common workaround of generating training data for multiple optimizer steps results in a lag between the currently trained policy and the behavior policy that generates the training data. The lagging off-policy data is known to harm the commonly used effective RL algorithms [Noukhovitch et al., 2024], including, REINFORCE [Williams, 1992], PPO [Schulman et al., 2017] and GRPO [Shao et al., 2024, Guo et al., 2025], because these algorithms were designed to be trained with on-policy or near on-policy data, with the behavior and current policy being very close.

In this paper, we present the PipelineRL approach to RL for LLMs that achieves a better trade-off between hardware utilization and on-policy learning. Like prior work on efficient RL [Espeholt et al., 2018, 2019], PipelineRL features concurrent asynchronous data generation and training. PipelineRL adapts prior asynchronous RL ideas to long-sequence generation with LLMs by introducing *in-flight weight updates*. As shown in Figure 1, during an in-flight weight update the LLM generation engine

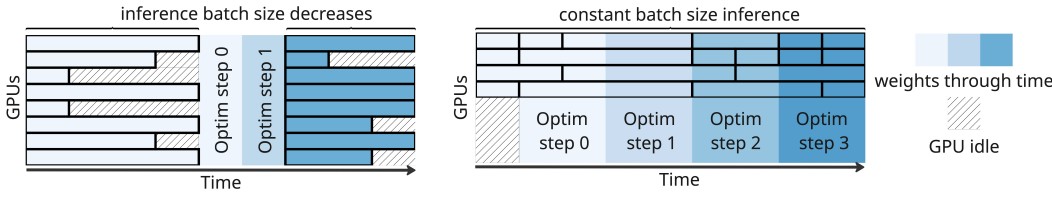

Figure 1: **a)** Conventional RL alternates between using all the GPUs for generation and then training. **b)** PipelineRL runs generation and training concurrently, always using the freshest model weights for generations thanks to the in-flight weight updates.

only briefly pauses to receive the model weights via a high-speed inter-accelerator network, and then proceeds to continue the generation of in-progress token sequences. In-flight updates eliminate the wasteful waits for the last sequence to finish, ensure high accelerator utilization at a constant generation batch size, and maximize the policy adherence of the recently generated tokens.

Our experiments on RL training for long-form reasoning show that on 4 DGX-H100 nodes, PipelineRL learns $\sim 2x$ faster than the comparable conventional RL baseline. We also observe that PipelineRL training data stays highly on-policy, and that models trained by PipelineRL perform comparably to similarly trained models from the literature. Lastly, a key contribution of this work is a scalable and modular PipelineRL implementation that we release as open-source software.[1]

## 2  Background

### 2.1  Reinforcement Learning for Large Language Models

Reinforcement learning (RL) is commonly used to train Large Language Models (LLM) to respect human preferences [Ouyang et al., 2022] for the LLM's outputs or to perform long-form reasoning to solve problems [Guo et al., 2025]. One can view LLM's weights as parameterizing a multi-step policy that assigns probabilities to the next token $y_i$ given the prompt $x$ and the previously generated tokens $y_{<i}$:

$$\pi(y|x) = \prod_{i=1}^{n} \pi(y_i|x, y_{<i}). \tag{1}$$

Recent works have shown that variations of basic policy gradient algorithms such as REIN-FORCE [Williams, 1992] are as effective for training LLMs as more sophisticated alternatives [Ahmadian et al., 2024, Roux et al., 2025]. Given a set of prompts $x_1, \ldots, x_m$, REINFORCE maximizes the expected return $J(\pi)$ of the policy $\pi$ by following an estimate $\tilde{\nabla} J(\pi)$ of the policy gradient $\nabla J(\pi)$:

$$J(\pi) = \frac{1}{m} \sum_{j=1}^{m} \left[ \mathbb{E}_{y \sim \pi(\cdot|x_j)} R(x_j, y) \right] \tag{2}$$

$$\nabla J(\pi) = \frac{1}{m} \sum_{j=1}^{m} \left[ \mathbb{E}_{y \sim \pi(\cdot|x_j)} \nabla \log \pi(y \mid x_j) R(x_j, y) \right] \tag{3}$$

$$\tilde{\nabla} J(\pi) = \frac{1}{mK} \sum_{j=1}^{m} \sum_{k=1}^{K} \nabla \log \pi(y \mid x_j) \left( R(x_j, y_k) - v_k(x_j) \right), \tag{4}$$

where $v_k(x_j)$ is the control variate term that reduces the estimate's variance, and $K$ is the number of samples per prompt $x$. In this study, we use the empirical mean $v_k(x_j) = \sum_{k=1}^{K} R(x_j, y_k)/K$ as the control variate.

In most practical RL setups, the *current policy* $\pi$ will often slightly differ from the *behavior policy* $\mu$ that generates $y_k$. This difference is usually handled by either a trust region constraint [Schulman

---

[1]The code is available online under Apache 2 license, we will add the link to the camera-ready version

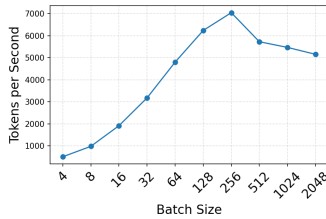 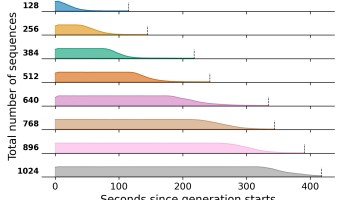 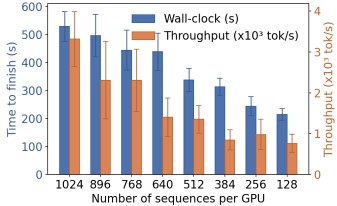

(a) Throughput vs batch size.  (b) Inference batch size vs time.  (c) Time vs Throughput.

Figure 2: **Analysis of generation times and throughput.** We perform all measurements using a vLLM engine serving a Qwen 2.5 7B model on a H100 GPU. **(a)** Short prompt generation throughput increases up to batch size 256. **(b)** Generation batch size gradually decreases to suboptimal values as the engine finishes sequences **(c)** Generation time reaches a plateau and throughput decreases as the number of sequences per GPU goes down. We report the average of 5 runs and 95% CI.

et al., 2017] or using Importance Sampling (IS). In practice, the importance weights are truncated to reduce the variance of the estimator [Munos et al., 2016, Espeholt et al., 2018]:

$$\tilde{\nabla}_{IS} J(\pi) = \frac{1}{mK} \min\left(c, \frac{\pi(y \mid x)}{\mu(y \mid x)}\right) \left(R(x_j, y_k) - v_k(x_j)\right) \nabla \log \pi(y \mid x). \qquad (5)$$

The Effective Sample Size (ESS) [Kong, 1992] is commonly used to quantify the quality of importance sampling estimators in RL [Schlegel et al., 2019, Fakoor et al., 2020]. When using off-policy RL, ESS measures how many samples from the current policy $\pi$ would yield equivalent performance to weighted samples from the behavior policy $\mu$. The (normalized) ESS is defined as:

$$\text{ESS} = \left(\sum_{i=1}^{N} w_i\right)^2 / N \sum_{i=1}^{N} w_i^2 \qquad (6)$$

where $w_i$ are importance weights for a sample of size $N$. This metric effectively ranges between 0 and 1 when normalized, with values closer to 1 indicating more efficient sampling, e.g. the ESS of on-policy data is exactly 1. Small ESS will result in a high variance REINFORCE gradient estimate and might destabilize the learning process.

## 2.2 Conventional RL

Most RL implementations alternate between generating sequences and training the policy on the generated data. We refer to this approach as Conventional RL and describe it in detail in Algorithm 1. When training involves doing $G > 1$ optimizer steps, the current policy $\pi$ gets ahead of the behavior policy $\mu$ that was used to generate the data. We adopt the term *lag* to refer to the number of optimizer steps between $\mu$ and $\pi$.

## 2.3 Efficient Sequence Generation with LLMs

Transformer models generate sequences one token at a time, left-to-right. To make this process efficient, advanced generation (inference) engines such as vLLM and SGLang process a batch of sequences at a time, while carefully managing their past keys and values in a paged structure called KV cache [Kwon et al., 2023]. All modern generation engines support adding new generation requests *in-flight* to the ones in progress without stopping the generation process. Based on accelerator specifications, generation engines should achieve the maximum generation throughput at very large batch sizes of several thousand sequences [2]. In practice, at very large batch sizes, the per-sequence latency can become prohibitively high, KV cache may grow too large to fit in accelerator memory, or the request queue management overheads can dominate.

---

[2]https://docs.nvidia.com/deeplearning/performance/dl-performance-matrix-multiplication/index.html

**Algorithm 1** Conventional RL

---

**Require:** Current policy $\pi$.
**Require:** Optimizer state opt_state.
**Require:** Number of optimizer steps per RL step $G$.
**Require:** Training batch size $B$.
  **while** True **do**
    **// generation**                                                                       ▷ RL step starts
    $\mu \leftarrow \pi$                               ▷ Initialize behavior policy $\mu$
    sequences $\leftarrow$ generate $BG$ sequences from $\mu$
    batches $\leftarrow$ split sequences in G batches of size B
    **// training**
    lag $\leftarrow 0$                                  ▷ lag between $\mu$ and $\pi$
    **for** batch **in** batches **do**
        $\pi$, opt_state $\leftarrow$ optimizer_step($\pi$, opt_state, batch)
        lag $\leftarrow$ lag + 1
    **end for**                                                          ▷ RL step ends
  **end while**

---

## 3   The learning speed ceiling of Conventional RL

Reinforcement learning for LLMs can be slow when the LLM is trained to generate long sequences of tokens, e.g., long-form reasoning to solve mathematical problems, because each generation can take up to several minutes. Here we explain why it is challenging to effectively scale up long sequence RL, i.e. to effectively use a larger number of accelerators $N$ to make average reward $R(t)$ at time $t$ grow faster. As a mathematical function, one can view $R(t)$ as a composition of the functions $R(S)$ and $S(t)$, where $S$ is the number of samples the RL learner will have processed by time $t$. A faster RL learner will have a higher *learning speed* $\frac{dR}{dt}$ which we can express as the product of *learning effectiveness* and *learning throughput* as follows:

$$\underbrace{\frac{dR}{dt}}_{\text{speed}} = \underbrace{\frac{dR}{dS}}_{\text{effectiveness}} \times \underbrace{\frac{dS}{dt}}_{\text{throughput}} . \tag{7}$$

The Conventional RL algorithm from Algorithm 1 has the highest $\frac{dR}{dS}$ when it is fully on-policy, i.e., when one performs only one optimizer step per each RL step. Yet the throughput $\frac{dS}{dt}$ in the pure on-policy case can be low because the accelerators will be working on at most batch size $B$ samples at a time. Increasing the number of accelerators $N$ will yield diminishing returns in increasing $\frac{dS}{dt}$, because the throughput of each accelerator will decrease when the number of samples per accelerator $\frac{B}{N}$ goes below the optimal range (Figure 2c). For example, see Figure 2a for inference throughput for a 7B Qwen model on a single H100 GPU. One can see that the throughput increases almost linearly up to the generation batch size of 128. Hence, e.g. using $2N$ GPUs to generate 32 samples will not be much faster than using $N$ GPUs to generate $64$. Furthermore, as the LLM finishes the shorter generations, there will be fewer longer generations still in progress, see Figure 2b for an illustration. Hence, to make good use of the hardware, one should use each accelerator to generate many times more sequences than the optimal batch size.

Commonly, to increase the throughput, most practitioners perform multiple $G > 1$ optimizer steps per RL step, which entails generating $BG$ rollouts at each generation stage. This way, one can often achieve a higher throughput $\frac{dS}{dt}$ by increasing $N$ up to a point when $\frac{BG}{N}$ becomes too small. It is, however, known from the literature that going too off-policy by using a high value of $G$ will eventually decrease the learning effectiveness $\frac{dR}{dN}$ [Noukhovitch et al., 2024]. Clearly, at some points, the rollouts from the old policy become too stale and no longer useful as the source of learning signal for the current policy. Hence, given a fixed optimizer batch size $B$, one scales up Conventional RL by increasing $G$ and $N$ until the product $\frac{dR}{dS}\frac{dS}{dt}$ no longer improves, and the hard ceiling of $\frac{dR}{dt}$ for the given number of accelerators $N$ is achieved.

---

**Algorithm 2** PipelineRL

---

**Require:** Current policy weights $\pi$.
**Require:** Generation batch size $H$.
**Require:** Training sequence queue $Q_{train}$.
 1: **function** ACTOR($\pi$)
 2:      sequences in progress $S_{prog} \leftarrow []$
 3:      **while** True **do**
 4:          $S_{fin}, S_{prog} \leftarrow$ pop finished sequences from $S_{prog}$
 5:          $Q_{train}.put(S_{fin})$                    ▷ Send finished seqs to the trainer
 6:          **if** $len(S_{prog} < H)$ **then**
 7:             add $H - len(S_{prog})$ prompts to $S_{prog}$
 8:          **end if**
 9:          **if** Trainer requests weight update **then**          ▷ In-flight check for new weights
10:             $\pi \leftarrow$ receive_weight_update()
11:             $\mu \leftarrow \pi$                              ▷ 0 lag between $\pi$ and $\mu$
12:          **end if**
13:          $S_{prog} \leftarrow$ generate next tokens with $\mu$
14:      **end while**
15: **end function**
16: **function** TRAINER($\pi$, opt_state)
17:      batch $\leftarrow []$
18:      **while** True **do**
19:          batch $\leftarrow$ get $B$ sequences from $Q_{train}$
20:          $ESS \leftarrow$ get_effective_sample_size($\pi$, batch)
21:          **if** ESS < threshold **then**
22:             sleep(until $Q_{train}$ contains on-policy data for $\pi$)
23:             continue
24:          **end if**
25:          $\pi$, opt_state $\leftarrow$ optimizer_step($\pi$, opt_state, batch)
26:          request_actor_weight_update($\pi$)             ▷ In-flight weight update
27:      **end while**
28: **end function**

---

## 4   Pushing the learning speed ceiling with PipelineRL

The Pipeline RL method differs from Conventional RL in two aspects: (1) running training and generation in parallel asynchronously, and (2) updating the generation weights after every optimizer step *in-flight*, i.e. without stopping the sequence generation. Algorithm 2 provides an abstracted formal description of PipelineRL in terms of two concurrent Actor and Trainer processes that communicate via a sample queue and a high-bandwidth weight transfer network.

The effectiveness-throughput trade-off for PipelineRL is the opposite of that of Conventional RL. Namely, adding more accelerators to a PipelineRL setup leads to a linear increase of $\frac{dS}{dt}$, but may eventually harm $\frac{dR}{dS}$. In Figure 3a, we illustrate how PipelineRL produces *mixed-policy sequences* in which earlier tokens are more off-policy than the recent ones. Doubling $N$ will double the lag of the earliest tokens as well as the average lag in the PipelineRL batch. Notably, the off-policyness profile is different for PipelineRL and its conventional counterpart. Taking the average token lag as a proxy for off-policyness, in PipelineRL all batches are equally off-policy, whereas for Conventional RL later batches become progressively more off-policy. This difference makes it hard to analytically reason about the $\frac{dR}{dt}$ improvement that PipelineRL can bring over the baseline, because $\frac{dR}{dS}$ can only be estimated empirically by running RL experiments. In supplementary material, we present our simulation of how, for the same maximum lag $g_{max}$ PipelineRL can learn 1.5x faster than Conventional RL. The empirical gains can be even larger, depending on how frequently one can make weight updates without hurting the learning effectiveness $\frac{dR}{dS}$.

**Configuring PipelineRL vs Conventional RL**     For a fixed batch size $B$ and a number of accelerators $N$, one can configure Conventional RL by choosing the number of optimizer steps $G$, trading off

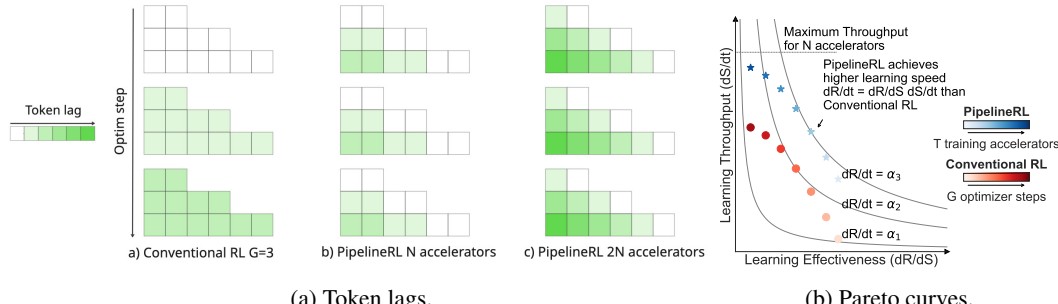

(a) Token lags.  (b) Pareto curves.

Figure 3: **(a)** For Conventional RL, the token lag increases with the number of optimizer steps. In PipelineRL with N accelerators, the token lag varies throughout the sequence, where earlier tokens have higher lag. The lag structure in each batch is the same. Doubling the PipelineRL accelerators, everything else constant, double the lag of early tokens. **(b)** Schematic illustration of PipelineRL's throughput-effectiveness trade-off as a function of training accelerators $T$ and of Conventional RL as a function of lag $G$. PipelineRL achieves a higher $\frac{dR}{dS}\frac{dS}{dt}$ for the same number $N$ of accelerators.

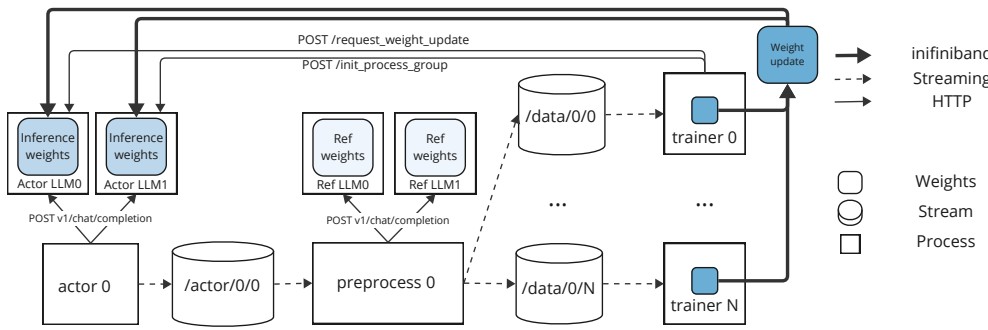

Figure 4: The three pipeline stages of PipelineRL implementation: actor, preprocessor and trainer. Earlier stages stream the data to the latter ones using Redis as the streaming broker.

the learning effectiveness for the throughput. The PipelineRL configuration can likewise be mostly reduced to a single parameter, namely the number of training accelerators $T$ out of $N$ available ones. Setting a higher $T$ will almost linearly decrease the time $t_{train}$ that is needed for the trainer to process $B$ sequences and perform an optimizer step. $T$ effectively determines the optimal generation batch size $H$ to be used at all $N - T$ accelerators. Using a lower $H$ leads to a lower maximum generation latency $t_{gen}$, which consequently reduces the maximum lag $g_{max} = \lceil t_{gen}/t_{train} \rceil$. Hence, it makes sense to use the smallest $H$ that suffices to produce enough training data. Consequently, the maximum lag $g_{max}$ for PipelineRL grows with the number of training accelerators $T$, as higher $T$ requires a higher $H$ and leads to a lower $t_{train}$ and a higher $t_{gen}$. On the contrary, the sample throughput of PipelineRL grows with $T$ up to a point when $N - T$ accelerators cannot generate enough data for the over-powered trainer. We recommend avoiding extreme configurations with $T$ too high (very high lag $G$) and $T$ too low (bad hardware utilization, one can just as well scale down the compute). Figure 3b visualizes how different configurations of PipelineRL and Conventional RL achieve different learning effectiveness $\frac{dR}{dS}$ and throughput $\frac{dS}{dt}$, with PipelineRL setups reaching higher $\frac{dR}{dt} = \frac{dS}{dt}\frac{dR}{dS}$ isocurves.

**PipelineRL Safety Mechanism**   While in-flight weight updates can be useful, on the flip side, the mixed-policy sequences generated by the in-flight behavior policy can present a risk to the stability of the training process, in particular because after an in-flight weight update, the generation server continues with the stale key and value vectors that were computed by a prior version of the model. To remediate these risks, we monitor the Effective Sample Size (ESS) of each training batch. Once ESS drops below a certain threshold, we stop updating the current policy until it accumulates a full batch of purely on-policy sequences, see lines 21-23 in Algorithm 2.

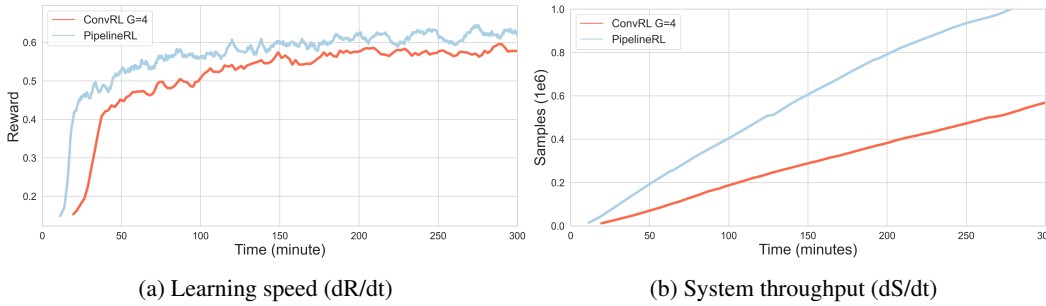

(a) Learning speed (dR/dt)  (b) System throughput (dS/dt)

Figure 5: **Learning speed and throughput.** PipelineRL achieves higher throughput and learning speed than Conventional RL with G=4 optimizer steps per each RL step.

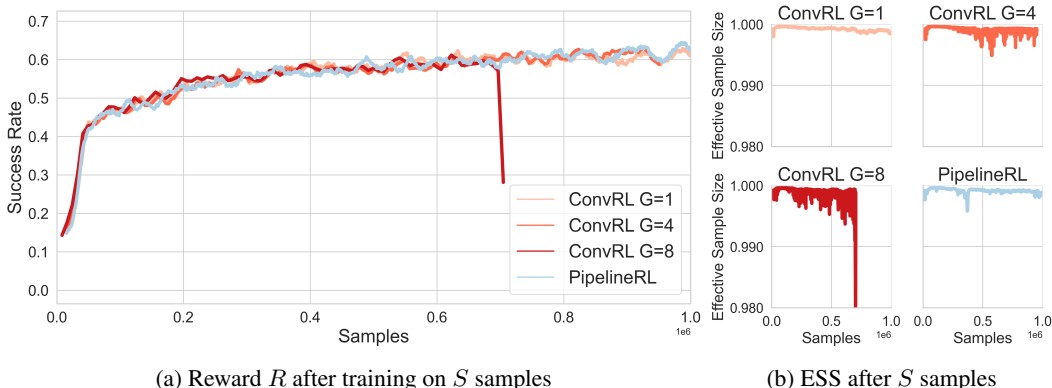

(a) Reward $R$ after training on $S$ samples  (b) ESS after $S$ samples

Figure 6: **(a)** PipelineRL attains the same average rewards for each number of training samples as pure on-policy $G = 1$ Conventional RL **(b)** PipelineRL stays mostly on-policy.

**Architecture and Implementation Details**  Our PipelineRL implementation concurrently runs many distributed vLLM generation engines and DeepSpeed training workers in a three stage pipeline that we describe in Figure 4. The middle Preprocessor stage that we omitted from Algorithm 2 for simplicity, computes reference model log-probabilities often used in Reinforcement Learning from Human Feedback [Ouyang et al., 2022]. The PipelineRL architecture is highly modular — any generation software that supports the three HTTP API endpoints that PipelineRL requires can be easily integrated in the future. The three APIs are the popular `/v1/chat/completions` for generation, `/init_process_group` for creating the weight transfer process group, and `/request_weight_update` for initiating the in-flight weight update. Key optimizations in PipelineRL include online sequence packing for fast training and using ring buffers to minimize the lag when earlier pipeline stages run faster than the later ones, e.g. when the trainer makes a checkpoint.

## 5  Experiment

For the experimental validation of PipelineRL's high learning effectiveness $\frac{dR}{dS}$ and throughput $\frac{dS}{dt}$, we have chosen the challenging task of training a base (i.e. not instruction-tuned) model to perform long-form reasoning to solve mathematical problems. We find this task to be a great testbed for PipelineRL because the policy undergoes rapid changes over the course of training. In particular, the length of generated sequences grows dramatically [Guo et al., 2025], making it essential to stay on-policy for effective learning.

**Experimental setup.**  For each experiment, we train the Qwen 2.5 base model [Yang et al., 2024] with 7B parameters on 17K math problems from the OpenReasoner Zero dataset [Hu et al., 2025] for 1000 optimizer steps with the batch size $B = 1024$. We use Adam optimizer [Kingma, 2014] with the learning rate 1e-6. We run the PipelineRL experiments on 4 DGX-H100 nodes, using 16 GPUs

for generation at batch size $H = 64$ and 16 GPUs for training. We tweak PipelineRL to simulate Conventional RL by accumulating and shuffling a buffer of $BG$ samples at the Preprocessor stage before the $G$ optimizer steps of each RL step start. To estimate the Conventional RL throughput, we use 4 nodes for generation at batch size $H = 128$ and 2 nodes for training, and then add a correction for training on 2x fewer GPUs than what an efficient Conventional RL implementation with a quick generation-training transition could use. We give reward 1 to any generated sequence with the correct answer and 0 otherwise. We train every model with importance weighted REINFORCE as described in Section 2 and clamp the importance weights to 5.

Table 1: Success rate of models trained with PipelineRL compared to results in the literature.

| Method | Math 500 | AIME24 | # samples ($\cdot 10^6$) | training data |
|---|---|---|---|---|
| Qwen 2.5 base 7b | 31.6 | 3.3 | - | - |
| SimpleRL Zero [Zeng et al., 2025] | 78.2 | 20.0 | 0.82 | Math Level 3-5 |
| OpenReasoner Zero [Hu et al., 2025] | $\sim 82.0$ | $\sim 20.0$ | 8.2 | OpenReasoner |
| PipelineRL (batch size 1024) | 81 | 17.5 | 2.0 | OpenReasoner |
| PipelineRL (batch size 4096) | 84.6 | 19.8 | 6.2 | OpenReasoner |

**PipelineRL learns faster due to higher throughput.** We compare the learning speed of PipelineRL to that of Conventional RL with $G = 4$ optimizer steps, as that was the maximum $G$ for which Conventional RL training was stable. PipelineRL achieves the same reward values approximately $\sim 2x$ faster than this baseline (Figure 5a) due to $\sim 2x$ faster sample throughput (Figure 5b). The main cause of the throughput increase is that GPU utilization for $G = 4$ experiment on 32 GPUs is relatively low for each GPU when it has to generate just 4096 / 32 = 256 sequences (see Figure 2b).

**PipelineRL learns effectively.** To better measure learning effectiveness $\frac{dR}{dS}$ of PipelineRL, we also run Conventional RL experiments with $G = 1$ and $G = 8$ optimizer steps. Notably, the $R(S)$ curves are indistinguishable for all compared methods up to a point when high G runs diverge, likely because of going too far off-policy. This result validates that PipelineRL's signature in-flight weight updates do no harm to the sequence generation process. For the PipelineRL run the ESS safety mechanism was never triggered, but in our preliminary experiments, it was sometimes activated and prevented the policy blow-up.

**PipelineRL matches comparable results on reasoning tasks.** Table 1 compares the test performance of PipelineRL to similar experiments that start training from the same Qwen 2.5 7B model. In this experiment we used batch size 4096 because we found it leads to a higher performance. On the math reasoning benchmarks MATH500 [Hendrycks et al., 2021] and AIME2024 [Li et al., 2024]. PipelineRL matches or exceeds the performance of Open Reasoner Zero and SimpleRL Zero.

**PipelineRL stays more on-policy.** To gain a better understanding of which training methods stay more on-policy, we plot the evolution of the ESS on-policyness measure throughout the training. Figure 6b shows that for a purely on-policy run with $G = 1$, ESS stays close to 1.[3] For $G = 8$, ESS generally decreases with the lag between the behavior and the current policy. We note that the magnitude of the ESS drop varies throughout training for $G = 4$ and $G = 8$ runs. The ESS of PipelineRL follows a different pattern. It stays close to ESS of $G = 1$ gold-standard run with some large drops when the current policy quickly shifts and the variance of the importance weights increases. These drops are the reason why we recommend using the ESS-based safety mechanism for PipelineRL. Notably, even though the maximum lag $g_{max}$ in our PipelineRL experiment was around 8 on average, Figure 6b shows that PipelineRL's ESS curves look more like that of $G = 1$ on-policy run than that of $G = 8$ more off-policy experiment. We believe it is due to the lag being lower than $g_{max}$ for a majority of tokens, since the average generated sequence length in our experiments ranged between 1K and 2K tokens, well below the 8K maximum.

---

[3]The reason for ESS falling below 0.999 for $G = 1$ is the consistent small difference between the log-probabilities produced by vLLM and Huggingface Transformers implementation of Qwen 2.5 model.

## 6   Related work

Asynchronous and high-throughput RL has been extensively studied. IMPALA [Espeholt et al., 2018] decoupled acting from learning to maximize GPU utilization. Like PipelineRL, IMPALA used truncated importance weights to estimate the value function from off-policy samples. Furthermore, IMPALA kept the policy weights constant for the length of an episode. SeedRL [Espeholt et al., 2019] proposed to update the model's parameters during an episode, resulting in trajectories where different actions were sampled by different policies. OpenAI Five [OpenAI et al., 2019] was trained using asynchronous PPO to achieve superhuman performance on Dota 2. These previous works were focused on RL for video games. Closer to our work, [Noukhovitch et al., 2024] explores asynchronous RL for LLMs. In their approach, data generation for the next $G$ optimizer steps is synchronized with training on the previous $G$ optimizer steps, leading to higher off-policyness than Conventional RL, unlike PipelineRL. The same study shows that offline methods such as DPO [Rafailov et al., 2023] can better tolerate off-policyness.

There exist several other scalable open-source RL implementations. veRL [Sheng et al., 2024] implements Conventional RL efficiently by using a sophisticated hybrid generation-training engine that supports quick transitions between training and generation on the same GPUs. We believe veRL's throughput would be similar to our Conventional RL baseline. Without the hybrid engine, in OpenRLHF [Hu et al., 2024] training GPUs idle during generation and vice-versa.

## 7   Conclusion and Discussion

We have shown how in-flight weight updates help PipelineRL break the learning speed ceiling of the conventional two-stage RL approach. We believe that for long sequence generation, in particular, this speedup would be very difficult to attain with another asynchronous RL approach, as synchronous waits for generation to finish would hurt the throughput and/or learning effectiveness. The stale KV-cache risk that in-flight updates introduce can be mitigated by recomputing the KV cache after each update, which can be done fast at a high GPU utilization, but will still lower the throughput.

We believe PipelineRL may be particular useful for training LLMs to excel at agentic behaviors that involve multiple LLM generations interspersed with environment interactions. Another promising direction for future work is to study when the recent low lag tokens in PipelineRL are helpful, and on the contrary, where PipelineRL's constantly high lag of early tokens in long sequences hurts.

**Limitations**   PipelineRL will only bring a limited throughput increase over Conventional RL if the LLM is asked to generate the exact same number of tokens for the same prompt. In this unlikely scenario, Conventional RL will be likewise capable of maintaining a constant generation batch size. The PipelineRL's stable average token lag and the low lag of recent tokens in each batch may, however, still affect the learning effectiveness. The PipelineRL throughput advantages will likewise decrease in setups with scarce or extensive compute resources. In the former case, each GPU will get enough generation tasks for the GPU utilization to be high. In the latter, the learning speed will be bounded not by the hardware utilization but by the best possible generation latency and by the environment feedback delay.

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
