# OpenReview forum: "PipelineRL: Faster On-policy Reinforcement Learning for Long Sequence Generation"
_NeurIPS.cc/2025/Conference — Submitted to NeurIPS 2025_

### Official Review · Reviewer_rF4g · 2025-07-02

**Clarity:** 3
**Significance:** 3
**Originality:** 4
**Rating:** 5
**Confidence:** 3

**Summary:**

This paper introduces PipelineRL, a novel asynchronous reinforcement learning approach designed specifically to improve efficiency when training large language models for tasks involving long sequence generation. The core innovation is in-flight weight update, which continuously updates model parameters asynchronously during ongoing sequence generation, addressing issues with GPU utilization and off-policyness. Experimental results show that PipelineRL achieves roughly 2x faster learning speeds and significantly higher GPU throughput compared to conventional RL methods. Additionally, the authors provide a modular, scalable, and open-source implementation to facilitate broader adoption.

**Questions:**

- Could the authors provide further insights into specific instability scenarios arising from continuous asynchronous updates? Are there recommended heuristics or additional safeguards beyond ESS-based monitoring?

- Have you explored or observed scalability limitations beyond the demonstrated 32-GPU setup? What are potential bottlenecks at larger scales?

- Could the authors provide additional guidance or insights into selecting optimal PipelineRL configurations (e.g., number of accelerators, optimal batch sizes, lag settings)?

- Could the authors discuss how PipelineRL’s performance and stability are influenced by variations in sequence length? Are there specific considerations or adjustments recommended when dealing with significantly shorter or longer sequences than those tested?

**Ethical Concerns:**

["NO or VERY MINOR ethics concerns only"]

**Final Justification:**

- **Resolved:**
  - Authors provided new experiments demonstrating scalability up to 128 GPUs, addressing concerns about limited scaling evidence.
  - KL-divergence analysis showed that inflight weight updates reduce or match divergence compared to conventional RL, mitigating concerns about instability and data quality.

- **Better to have:**
  - Comprehensive sensitivity analysis, study the robustness to hyperparameter settings.
  - Sequence length analysis is theoretical only, better to have empirical validation for very short or very long sequences.

- **Weighting:**
  - The main concerns around scalability (up to 128 GPUs) and stability have been adequately addressed, which strengthens the paper’s contribution.
  - Remaining issues are secondary but limit confidence in general applicability.

**Overall:** The rebuttal improved confidence in the paper’s main claims and I recommend to accept. However, it should be better to add the missing parts, so I keep my origin score.

**Limitations:**

- PipelineRL's substantial throughput gains depend heavily on hardware-specific infrastructure, such as high-bandwidth GPU interconnects and optimized GPU setups, potentially affecting its performance portability across varying hardware environments.

- Continuous asynchronous updates, despite ESS monitoring, could pose risks to training stability, particularly if hyperparameters or system configurations deviate significantly from tested settings.

**Paper Formatting Concerns:**

None.

**Quality:**

4

**Strengths And Weaknesses:**

## Strengths
- PipelineRL addresses critical RL bottlenecks (e.g., GPU utilization, maintaining on-policy data freshness) through innovative asynchronous updates, significantly improving training efficiency.

- Clear and comprehensive experiments convincingly demonstrate significant throughput improvements and learning efficiency relative to conventional RL approaches. Metrics such as Effective Sample Size (ESS) provide additional insights into the method’s effectiveness.

- Methodological details are supported by clear pseudocode and detailed justifications, enhance reproducibility and facilitate understanding.

- Releasing a scalable, modular open-source implementation significantly boosts practical utility and the potential impact of the paper.

## Weaknesses
- The continuous "in-flight weight updates" approach could lead to training instabilities, especially since behavior policies partly depend on outdated cached activations (KV vectors). Although the authors propose an ESS-based monitoring safeguard, it's unclear how comprehensively this mitigates instability under rapid or extensive model updates.

- While scalability is demonstrated convincingly up to 32 GPUs, the paper does not discuss potential scalability limitations at larger scales (e.g., hundreds of GPUs). Issues such as communication bottlenecks, KV-cache complexity, or diminishing returns in throughput at very large scales are plausible and warrant further investigation.

- PipelineRL’s performance improvements hinge on carefully tuned hyperparameters (e.g., ratio of training to generation GPUs, lag, batch sizes). The paper lacks a comprehensive sensitivity analysis, making it difficult to assess how robust the method is to variations in these critical parameters.

- Since PipelineRL specifically targets long-sequence generation, it would be valuable to see analysis or discussion on how variations in sequence lengths affect PipelineRL’s throughput, effectiveness, and stability.

---

> ### Author Rebuttal · Authors · 2025-07-31
>
> Thank you for your insightful review. We first highlight our new experiments and then addresses your comments one by one.
>
> We provide new experiments to addresses 1) scalability, and 2) data quality and stability.
>
> First, we improve the stability fo the code base by adding a value function. The value function resulted in Conventional RL being stable up to G=16 from G=4, making it a much stronger baseline. Furthermore, we scaled PipelineRL to 128 GPUs and still observe a significant throughput improvement over the baseline.
>
> | Method | Model | GPUs | Tokens Generated & Trained | Training Time | Throughput Improvement | Train Performance |
> |--------|-------|------|----------------------------|---------------|----------------------|-------------------|
> | **PipelineRL** | Qwen 2.5 Base 7B | 128 | ~1B tokens | ~75 minutes | **2.2x faster** | 0.56 |
> | **Conventional RL (G=16)** | Qwen 2.5 Base 7B | 128 | ~1B tokens | ~165 minutes | Baseline | 0.56 |
>
> Second, in order to understand the impact of inflight weight update, we compute the KL with the target policy ($\pi$) for different max lag for the conventional RL behavior policy ($\pi_C$) and PipelineRL behavior policy ($\mu$) for 3 different checkpoints. Interestingly, we oberve that PipelineRL generally obtains lower KL divergence between the behavior and target policy for max lag greater than 1.
>
> |                                                                    | $g_{\max}$=1   | $g_{\max}$=4   | $g_{\max}$=16   | $g_{\max}$=32   |
> |:-------------------------------------------------------------------|:---------------|:---------------|:----------------|:----------------|
> | **Starting checkpoint C at optimizer step 0**                      |                |                |                 |                 |
> | Fully Off-policy: KL $(\pi_C \Vert \pi)$                           | **3.98e-07**   | **4.29e-07**   | 1.06e-03        | 5.46e-03        |
> | Mixed-policy: KL $(\mu \Vert \pi)$                                 | 2.82e-06       | 4.03e-06       | **6.19e-04**    | **2.04e-03**     |
> | **Starting checkpoint C at optimizer step 100**                    |                |                |                 |                 |
> | Fully Off-policy: KL $(\pi_C \Vert \pi)$                           | **4.72e-06**       | 2.57e-05       | 3.71e-04     | 1.90e-03        |
> | Mixed-policy: KL $(\mu \Vert \pi)$                                 | 5.77e-06       | **1.90e-05**    | **1.96e-04**    | **6.14e-04**    |
> | **Starting checkpoint C at optimizer step 190**                    |                |                |                 |                 |
> | Fully Off-policy: KL $(\pi_C \Vert \pi)$                           | **2.35e-05**       | 1.81e-04       | 2.86e-03        | 6.89e-03        |
> | Mixed-policy: KL $(\mu \Vert \pi)$                                 | 3.01e-05       | **1.53e-04**       | **2.01e-03**        | **2.86e-03**        |
>
> ------
>
> > The continuous "in-flight weight updates" approach could lead to training instabilities, especially since behavior policies partly depend on outdated cached activations (KV vectors). Although the authors propose an ESS-based monitoring safeguard, it's unclear how comprehensively this mitigates instability under rapid or extensive model updates.
>
> In order to assess the data quality produced by PipelineRL, we added an experiment (see Exp 2 above) to compute the KL divergence between the policy with inflight weight update and the target policy. We observe that the inflight weight update policy obtains similar or lower KL divergence than Conventional RL for the same maximum lag. This implies that weight updates can improve the on-policiness of the collected data and its quality.
>
>
> > While scalability is demonstrated convincingly up to 32 GPUs, the paper does not discuss potential scalability limitations at larger scales (e.g., hundreds of GPUs). Issues such as communication bottlenecks, KV-cache complexity, or diminishing returns in throughput at very large scales are plausible and warrant further investigation.
>
> We successfully scaled PipelineRL to 128 GPUs and show significant speed improvement (see Exp 1 above). As mentionned in Figure 3, the major concern in scaling up the number of GPUs is the max lag will increase and can potentially destabilize training.
>
> > PipelineRL's performance improvements hinge on carefully tuned hyperparameters (e.g., ratio of training to generation GPUs, lag, batch sizes). The paper lacks a comprehensive sensitivity analysis, making it difficult to assess how robust the method is to variations in these critical parameters.
>
> In our experience, allocating 50% of your GPUs for training and 50% for inference is a good starting point. One can then monitor if the trainer is starved or overrun with data and adjust the allocation accordingly.
>
> > Since PipelineRL specifically targets long-sequence generation, it would be valuable to see analysis or discussion on how variations in sequence lengths affect PipelineRL's throughput, effectiveness, and stability.
>
> * Please see appendix A3, for a theoretical derivation of how the max sequence length impact the lag and constain throughput and appendix A4 for a theoretical speed up computation.
>
>
> **Questions:**
> > Could the authors provide further insights into specific instability scenarios arising from continuous asynchronous updates? Are there recommended heuristics or additional safeguards beyond ESS-based monitoring?
>
> In order to assess the potential instability caused by using inflight weight updates, we added an experiment (see Exp 2 above) to compute the KL divergence between the policy with inflight weight update and the target policy. We observe that the inflight weight update policy obtains similar or lower KL divergence than Conventional RL for the same maximum lag. This implies that weight updates can improve the on-policiness of the collected data and its quality.
>
> > Have you explored or observed scalability limitations beyond the demonstrated 32-GPU setup? What are potential bottlenecks at larger scales?
>
> Please see experiment 1 above. We successfully scaled PipelineRL to 128 GPUs.
>
> > Could the authors provide additional guidance or insights into selecting optimal PipelineRL configurations (e.g., number of accelerators, optimal batch sizes, lag settings)?
>
> Please see Appendix A3 for a dicussion of PipelineRL as a function of accelerators, batch size, and the resulting lag.
>
> > Could the authors discuss how PipelineRL's performance and stability are influenced by variations in sequence length? Are there specific considerations or adjustments recommended when dealing with significantly shorter or longer sequences than those tested?
>
> On the one hand, for very short sequences, e.g. 1 token, PipelineRL would not provide much improvement over Conventional RL (for high enough G) since the inference GPU would also run at full batch size. On the other hand, for very long sequences, e.g. 1M tokens, the performance gains over conventional RL will be large, but the max lag will increase and could potentially destabilize learning. However, lowering the learning rate, increasing the batch size, and excluding tokens with very high lag could potentially be used to stabilize training and still obtain good throughput.

---

> > ### Comment · Reviewer_rF4g · 2025-08-07
> >
> > Thank you for the detailed and thoughtful rebuttal, as well as conducting new experiments. My concerns are resolved.

---

### Official Review · Reviewer_BJb5 · 2025-07-02

**Clarity:** 3
**Significance:** 3
**Originality:** 3
**Rating:** 5
**Confidence:** 2

**Summary:**

This paper introduces PipelineRL framework to improve the efficiency of LLM training on long context tasks. It identifies scalability limitation in conventional RL and put forward a pipelined architecture that runs generation and training in parallel, by introducing in-flight weight updates. This allows the training to proceed with updated policies without pausing generation. The evaluation results showcase $2\times$ faster training compared to conventional RL. The framework seems to be open-sourced and tested on math reasoning tasks.

**Questions:**

- Do you anticipate similar performance improvements when applying PipelineRL to other domains like code generation and dialog?
- Is there any plans to perform ablations to isolate the contribution of in-flight updates vs. asynchronous pipeline parallelism to the overall speedup?

**Ethical Concerns:**

["NO or VERY MINOR ethics concerns only"]

**Final Justification:**

I thank the authors to have engaged with reviewers' comments.

My original comments were addressed textually:
1. Experiments are limited to Mathmatical and what's anticipated in terms of performance generalisation to other tasks: Authors answered this has to do with various sequence length in a given task.
2. Statistical variations: authors responded that they ran the experiments 3 times and in the process of another 2. This is not a critical point, in my opinion.

**Limitations:**

yes

**Quality:**

3

**Strengths And Weaknesses:**

Strengths

- The idea of in-flight weight update enables asynchronous generation and training, which seems novel and effective in maximizing GPU usage.
- The x2 improvement of training speed over conventionalRL is impressive.
- The authors have open-sourced the implementation of PipelineRL.
- The paper is well structured, clearly written and easy to follow.

Weakness

- Testing on mathematical reasoning model alone seems to be slightly underwhelming for the potential impact of this work. I am wondering if it including other tasks e.g. code generation would help to see generalisation of PipelineRL to other tasks?
- It would be helpful to include results from multiple runs and show variations e.g. standard deviation, to support the current results with statistical significance.

---

> ### Author Rebuttal · Authors · 2025-07-31
>
> Thank you for your review. We first highlight our new experiments and then addresses your comments one by one.
>
> We provide new experiments to addresses 1) scalability, and 2) data quality and stability.
>
> First, we improve the stability fo the code base by adding a value function. The value function resulted in Conventional RL being stable up to G=16 from G=4, making it a much stronger baseline. Furthermore, we scaled PipelineRL to 128 GPUs and still observe a significant throughput improvement over the baseline.
>
> | Method | Model | GPUs | Tokens Generated & Trained | Training Time | Throughput Improvement | Train Performance |
> |--------|-------|------|----------------------------|---------------|----------------------|-------------------|
> | **PipelineRL** | Qwen 2.5 Base 7B | 128 | ~1B tokens | ~75 minutes | **2.2x faster** | 0.56 |
> | **Conventional RL (G=16)** | Qwen 2.5 Base 7B | 128 | ~1B tokens | ~165 minutes | Baseline | 0.56 |
>
> Second, in order to understand the impact of inflight weight update, we compute the KL with the target policy ($\pi$) for different max lag for the conventional RL behavior policy ($\pi_C$) and PipelineRL behavior policy ($\mu$) for 3 different checkpoints. Interestingly, we oberve that PipelineRL generally obtains lower KL divergence between the behavior and target policy for max lag greater than 1.
>
> |                                                                    | $g_{\max}$=1   | $g_{\max}$=4   | $g_{\max}$=16   | $g_{\max}$=32   |
> |:-------------------------------------------------------------------|:---------------|:---------------|:----------------|:----------------|
> | **Starting checkpoint C at optimizer step 0**                      |                |                |                 |                 |
> | Fully Off-policy: KL $(\pi_C \Vert \pi)$                           | **3.98e-07**   | **4.29e-07**   | 1.06e-03        | 5.46e-03        |
> | Mixed-policy: KL $(\mu \Vert \pi)$                                 | 2.82e-06       | 4.03e-06       | **6.19e-04**    | **2.04e-03**     |
> | **Starting checkpoint C at optimizer step 100**                    |                |                |                 |                 |
> | Fully Off-policy: KL $(\pi_C \Vert \pi)$                           | **4.72e-06**       | 2.57e-05       | 3.71e-04     | 1.90e-03        |
> | Mixed-policy: KL $(\mu \Vert \pi)$                                 | 5.77e-06       | **1.90e-05**    | **1.96e-04**    | **6.14e-04**    |
> | **Starting checkpoint C at optimizer step 190**                    |                |                |                 |                 |
> | Fully Off-policy: KL $(\pi_C \Vert \pi)$                           | **2.35e-05**       | 1.81e-04       | 2.86e-03        | 6.89e-03        |
> | Mixed-policy: KL $(\mu \Vert \pi)$                                 | 3.01e-05       | **1.53e-04**       | **2.01e-03**        | **2.86e-03**        |
>
> ------
>
> > Testing on mathematical reasoning model alone seems to be slightly underwhelming for the potential impact of this work. I am wondering if it including other tasks e.g. code generation would help to see generalisation of PipelineRL to other tasks?
>
> Training a model from scratch to improve mathematical reasoning is particularly difficult since the sequence lengths vary acros problems and throughout training. The improvement will translate to other domains also having variablility in sequence lengths.
>
> > It would be helpful to include results from multiple runs and show variations e.g. standard deviation, to support the current results with statistical significance.
>
> Yes, we have ran 3 seeds for experiment 1 and are in the process of running 2 more seeds.
>
> > Do you anticipate similar performance improvements when applying PipelineRL to other domains like code generation and dialog?
>
> PipelineRL improvement over conventional RL are partly a function of the variance in sequence length. Therefore, a code/dialog dataset that results in sequence of different length would also benefit from PipelineRL.
>
> > Is there any plans to perform ablations to isolate the contribution of in-flight updates vs. asynchronous pipeline parallelism to the overall speedup?
>
> Removing in flight weight update would result in stopping the inference GPU and result in drastic slowdown of PipelineRL.

---

> > ### Comment · Reviewer_BJb5 · 2025-08-06
> >
> > Thank you for the response. I will keep my original rating.

---

### Official Review · Reviewer_ehFX · 2025-07-03

**Clarity:** 3
**Significance:** 3
**Originality:** 4
**Rating:** 4
**Confidence:** 3

**Summary:**

The manuscript introduces PipelineRL, a novel and impactful framework for RL of LLMs. Relative to existing asynchronous RL approaches, PipelineRL’s main contribution is in-flight weight updating, which promotes high accelerator utilization and enhances on-policyness of the generated data. Beyond demonstrating the efficacy and foundational rigor of the proposed algorithm, this work offers a scalable and extensible PipelineRL implementation.

**Questions:**

The following are listed in order of increasing importance. Addressing both Numbers 2 and 3, or addressing one aspect of Number 4 (which has 2 suggestions/aspects) would cause my score to increase to 5 (assuming other reviewers don't surface concerns I missed). Addressing more things could improve my score further.

1) Just curious: In Figure 5b, does the red line shift closer to the blue if G=8? I.e., can the speed gap be largely addressed by increasing G, and the main concern is how off-policy you become with large G?

2) Minor clarity concerns:
   - In Figure 2, I think “Generation time reaches a plateau and throughput decreases” should be flipped: in the plot, seemingly, generation time decreases and throughput decreases then levels off. Relatedly, consider adding text to Section 3 to clarify the implication of Figure 2b.
   - I think Equations 4 and 5 should have a “k” subscript for each occurrence of “y”. Equation 5 seems to be missing summation notation. Equation 6 seems to be missing parentheses around the divisor.
   - Line 126 and Figure 3a’s caption: stating some assumptions could help clarify when this analysis holds. E.g., I think the analysis assumes that N is below the value at which the queue Q is always sufficiently full when a sample of B sequences is requested, or assumes that training process throughput could be improved with more N. In other words, it seems like you could increase N and have no effect on S(t) for sufficiently large N.
   - Add training time to Table 1.

3) Major clarity concern. Consider adding more clarity around when PipelineRL provides benefits (in terms of sequence lengths, GPU counts, etc.). For example, what are the speedups as a function of max sequence length?

4) Baseline choice concern. I think there are two options to address this concern, pursuing both is not necessary but could be nice. The first idea may not be relevant given PipelineRL's focus on long context scenarios and the focus of prior work on shorter contexts.
   - Compare PipelineRL's speed and performance to those of asynchronous RL approaches' using the same models/data as the baselines, e.g. Async DPO (Noukhovitch et al., 2025) and TBA (Bartoldson et al., 2025). Relatedly, on line 34, consider referencing Async DPO and TBA as prior work that "features concurrent asynchronous data generation and training” for LLM RL.
   - Demonstrate the effectiveness of PipelineRL with another model (beyond Qwen) to illustrate the robustness of the proposed approach to different base models.

**Ethical Concerns:**

["NO or VERY MINOR ethics concerns only"]

**Final Justification:**

I would like to raise my score from 4 to 4.5, as the authors partially addressed my concerns. Right now, I have left my score as 4, but I am happy to change to 5 if the AC thinks that makes more sense.

Some concerns I raised that perhaps could have been better addressed include:
- tests of their method on a model other than Qwen
   - Qwen is known to respond positively to even incorrect rewards. I raised this point, but the authors were not interested in testing on another model. I suggested mentioning this as a limitation, then, and the authors were willing to do this (but perhaps should go further than their suggested statement).
- a need for more quantitative (i.e. experiment) or qualitative (i.e. discussion) comparisons to preexisting async RL approaches that speed up training ("async RLHF", "trajectory balance with asynchrony")
- various minor clarity concerns

Thanks!

**Limitations:**

Yes.

**Quality:**

4

**Strengths And Weaknesses:**

The paper is very high quality and mostly clearly written (some details might need clarification, see “Questions”). E.g., the framing of RL speed given in Equation 7 is great! Figure 3 shows the off-policyness/lag effect of conventional RL and PipelineRL nicely, and its speed comparison is helpfully illustrative.

The work is original and will be impactful for the community. E.g., its experiments clarify its throughput benefits, and demonstrate the high quality of samples produced in an in-flight-weight-updating context (relative to samples produced using a conventional approach). Moreover, it uses effective sample size to ensure high-quality optimization steps in an asynchronous RL framework.

There are no major weaknesses, but significance could be improved by demonstrating the method on more models, in more environments, and against more baselines. Significance could also be improved by further clarifying the types of regimes in which the proposed approach is expected to be effective.

---

> ### Author Rebuttal · Authors · 2025-07-31
>
> Thank you for your review. We first highlight our new experiments and then addresses your comments one by one.
>
> We provide new experiments to addresses 1) scalability, and 2) data quality and stability
>
> First, we improve the stability fo the code base by adding a value function. The value function resulted in Conventional RL being stable up to G=16 from G=4, making it a much stronger baseline. Furthermore, we scaled PipelineRL to 128 GPUs and still observe a significant throughput improvement over the baseline.
>
> | Method | Model | GPUs | Tokens Generated & Trained | Training Time | Throughput Improvement | Train Performance |
> |--------|-------|------|----------------------------|---------------|----------------------|-------------------|
> | **PipelineRL** | Qwen 2.5 Base 7B | 128 | ~1B tokens | ~75 minutes | **2.2x faster** | 0.56 |
> | **Conventional RL (G=16)** | Qwen 2.5 Base 7B | 128 | ~1B tokens | ~165 minutes | Baseline | 0.56 |
>
> Second, in order to understand the impact of inflight weight update, we compute the KL with the target policy ($\pi$) for different max lag for the conventional RL behavior policy ($\pi_C$) and PipelineRL behavior policy ($\mu$) for 3 different checkpoints. Interestingly, we oberve that PipelineRL generally obtains lower KL divergence between the behavior and target policy for max lag greater than 1.
>
> |                                                                    | $g_{\max}$=1   | $g_{\max}$=4   | $g_{\max}$=16   | $g_{\max}$=32   |
> |:-------------------------------------------------------------------|:---------------|:---------------|:----------------|:----------------|
> | **Starting checkpoint C at optimizer step 0**                      |                |                |                 |                 |
> | Fully Off-policy: KL $(\pi_C \Vert \pi)$                           | **3.98e-07**   | **4.29e-07**   | 1.06e-03        | 5.46e-03        |
> | Mixed-policy: KL $(\mu \Vert \pi)$                                 | 2.82e-06       | 4.03e-06       | **6.19e-04**    | **2.04e-03**     |
> | **Starting checkpoint C at optimizer step 100**                    |                |                |                 |                 |
> | Fully Off-policy: KL $(\pi_C \Vert \pi)$                           | **4.72e-06**       | 2.57e-05       | 3.71e-04     | 1.90e-03        |
> | Mixed-policy: KL $(\mu \Vert \pi)$                                 | 5.77e-06       | **1.90e-05**    | **1.96e-04**    | **6.14e-04**    |
> | **Starting checkpoint C at optimizer step 190**                    |                |                |                 |                 |
> | Fully Off-policy: KL $(\pi_C \Vert \pi)$                           | **2.35e-05**       | 1.81e-04       | 2.86e-03        | 6.89e-03        |
> | Mixed-policy: KL $(\mu \Vert \pi)$                                 | 3.01e-05       | **1.53e-04**       | **2.01e-03**        | **2.86e-03**        |
>
> ------
>
> > There are no major weaknesses, but significance could be improved by demonstrating the method on more models, in more environments, and against more baselines.
>
> Training a model from scratch to improve mathematical reasoning is particularly difficult since the sequence lengths vary acros problems and throughout training. The improvement will translate to other domains also having variablility in sequence lengths and other autoregressive models.
>
> > Significance could also be improved by further clarifying the types of regimes in which the proposed approach is expected to be effective.
>
> We provided extensive discussion of the throughput and lag of PipelineRL as a function of max sequence length, batch size, and number of accelerators in Appendix A3.
>
> > 1. Just curious: In Figure 5b, does the red line shift closer to the blue if G=8? I.e., can the speed gap be largely addressed by increasing G, and the main concern is how off-policy you become with large G?
>
> Yes, this is what we refer to as the `trade-off between hardware efficiency and data on-policyness for LLM training`. Increasing G improves hardware efficiency at the cost of more off-policy data.
>
> >    - Line 126 and Figure 3a's caption: stating some assumptions could help clarify when this analysis holds
>
> Figure 3 is a didactic example assuming that doubling the number of accelerators will double the throughput of the inference GPUs and of the training GPUs. Under these assumptions, for a sequence of length L, the lag of the earliest token will double.
>
> >    - Add training time to Table 1
>
> We will add the training time for our models.
>
> > 3. Major clarity concern. Consider adding more clarity around when PipelineRL provides benefits (in terms of sequence lengths, GPU counts, etc.). For example, what are the speedups as a function of max sequence length?
>
> * Please see appendix A3, for a theoretical derivation of how the max sequence length impact the lag and constrain throughput and appendix A4 for a theoretical speed up computation.
>
> > 4. Baseline choice concern:
> >    - Compare PipelineRL's speed and performance to those of asynchronous RL approaches
>
> In generation-bound RL scenarios like math reasoning, where sequence generation take significantly longer than training, async (on-policy) RL results in highly inefficient training GPUs utilization. The efficiency of async RL can be improved by using off-policy methods, i.e. training several time on the same data. In contrast, Conventional RL takes better advantage of the GPUs by using all the GPUs for either training or generating without the need to training several time on the same data.
>
> >    - Demonstrate the effectiveness of PipelineRL with another model (beyond Qwen)
>
> Qwen is currently the most popular model to train from scratch for math reasoning. Benchmarking PipelineRL on training Qwen from scatch makes it easy for the community to compare and evaluate our results.

---

> > ### Comment · Reviewer_ehFX · 2025-08-05
> >
> > Thank you for the responses! I am inclined to ultimately raise my score but would like to continue discussion in the meantime.
> >
> > > Qwen is currently the most popular model to train from scratch for math reasoning. Benchmarking PipelineRL on training Qwen from scatch makes it easy for the community to compare and evaluate our results.
> >
> > Yes, I wasn't making an argument for replacing your Qwen results. My suggestion is to supplement your Qwen results. Qwen has different RL dynamics than other models. Would it not be a more rigorous test of your approach to investigate it on a second model? If you agree, maybe this could be added as a limitation?

---

> > > ### Author Response · Authors · 2025-08-05
> > > **Added limitations**
> > >
> > > Thabnk you for your response. We will add this line to the limitation section:
> > >
> > > Our experiments are limited to Qwen models which are widely used in RL Research.

---

### Official Review · Reviewer_cc95 · 2025-07-03

**Clarity:** 3
**Significance:** 3
**Originality:** 3
**Rating:** 5
**Confidence:** 3

**Summary:**

The paper introduces PipelineRL, a novel approach to on-policy RL designed to enhance the training efficiency of LLMs during long-sequence generation tasks. Specifically, PipelineRL employs concurrent asynchronous data generation and model training, which allows updated model weights to be applied without interrupting token generation. This approach achieves superior hardware utilization and maintains data freshness, addressing the challenges of scaling RL methods for LLMs. Experiments on 32 H100 GPUs show that PipelineRL achieves ~2x faster learning compared to conventional RL methods while preserving on-policy data quality. The paper also provides a modular open-source implementation, demonstrating state-of-the-art performance on reasoning benchmarks like MATH500 and AIME2024, and highlights the scalability and stability of the method.

**Questions:**

See above

**Ethical Concerns:**

["NO or VERY MINOR ethics concerns only"]

**Final Justification:**

PipelineRL presents a compelling and timely contribution that advances the state of practice in large-scale RL fine-tuning for LLMs. The paper addresses a bottleneck in balancing GPU utilization with on-policy data freshness and proposes a principled mechanism to reconcile these competing objectives through in-flight weight updates.

My only reservation is not a blocker: the work would be strengthened by explicitly quantifying any potential degradation in data quality during training (e.g., shifts in return distributions, KL to the current policy at collection time, off-policy bias metrics, or downstream task performance deltas as staleness varies). Including such measurements would provide a more complete view of the trade-offs introduced by PipelineRL and help practitioners tune the system to their quality–throughput targets.

Overall, the paper’s method is straightforward, and the contributions are significant. I recommend acceptance.

**Limitations:**

yes

**Quality:**

3

**Strengths And Weaknesses:**

**Strengths:**

1. Core Contribution: The primary contribution of PipelineRL lies in its ability to optimize the trade-off between hardware utilization and on-policy data generation, which is crucial for efficient LLM training.
2. Relevance: The paper addresses a highly significant issue for both the LLM and RL communities—balancing throughput with on-policy data quality.
3. Technical Innovation: The work effectively demonstrates how in-flight weight updates can maximize GPU utilization while ensuring data freshness, showcasing a practical and impactful design.

**Weakness:**

While I find no major reasons to reject this paper, I have one suggestion: it would be valuable to include measurements of potential degradation in data quality during the PipelineRL training process. This would provide additional insights into the trade-offs introduced by the proposed approach.

---

> ### Author Rebuttal · Authors · 2025-07-31
>
> Thank you for your review.
>
> > it would be valuable to include measurements of potential degradation in data quality during the PipelineRL training process. This would provide additional insights into the trade-offs introduced by the proposed approach.
>
> In order to understand the impact of inflight weight update, we compute the KL with the target policy ($\pi$) for different max lag for the conventional RL behavior policy ($\pi_C$) and PipelineRL behavior policy ($\mu$) for 3 different checkpoints. Interestingly, we oberve that PipelineRL generally obtains lower KL divergence between the behavior and target policy for max lag greater than 1.
>
> |                                                                    | $g_{\max}$=1   | $g_{\max}$=4   | $g_{\max}$=16   | $g_{\max}$=32   |
> |:-------------------------------------------------------------------|:---------------|:---------------|:----------------|:----------------|
> | **Starting checkpoint C at optimizer step 0**                      |                |                |                 |                 |
> | Fully Off-policy: KL $(\pi_C \Vert \pi)$                           | **3.98e-07**   | **4.29e-07**   | 1.06e-03        | 5.46e-03        |
> | Mixed-policy: KL $(\mu \Vert \pi)$                                 | 2.82e-06       | 4.03e-06       | **6.19e-04**    | **2.04e-03**     |
> | **Starting checkpoint C at optimizer step 100**                    |                |                |                 |                 |
> | Fully Off-policy: KL $(\pi_C \Vert \pi)$                           | **4.72e-06**       | 2.57e-05       | 3.71e-04     | 1.90e-03        |
> | Mixed-policy: KL $(\mu \Vert \pi)$                                 | 5.77e-06       | **1.90e-05**    | **1.96e-04**    | **6.14e-04**    |
> | **Starting checkpoint C at optimizer step 190**                    |                |                |                 |                 |
> | Fully Off-policy: KL $(\pi_C \Vert \pi)$                           | **2.35e-05**       | 1.81e-04       | 2.86e-03        | 6.89e-03        |
> | Mixed-policy: KL $(\mu \Vert \pi)$                                 | 3.01e-05       | **1.53e-04**       | **2.01e-03**        | **2.86e-03**        |
>
>
> We also improve the stability fo the code base by adding a value function. The value function resulted in Conventional RL being stable up to G=16 from G=4, making it a much stronger baseline. Furthermore, we scaled PipelineRL to 128 GPUs and still observe a significant throughput improvement over the baseline.
>
> | Method | Model | GPUs | Tokens Generated & Trained | Training Time | Throughput Improvement | Train Performance |
> |--------|-------|------|----------------------------|---------------|----------------------|-------------------|
> | **PipelineRL** | Qwen 2.5 Base 7B | 128 | ~1B tokens | ~75 minutes | **2.2x faster** | 0.56 |
> | **Conventional RL (G=16)** | Qwen 2.5 Base 7B | 128 | ~1B tokens | ~165 minutes | Baseline | 0.56 |

---

> > ### Comment · Reviewer_cc95 · 2025-08-03
> > **Comments after Author Discussion**
> >
> > Thank you for your response. I have no additional questions.

---

### Decision · Program_Chairs · 2025-09-17

**Decision:**

Reject

**Comment:**

This paper proposes PipelineRL, a framework that leverages in-flight weight updates to balance GPU utilization and on-policy data freshness during RL with LLMs. Reviewers appreciated the clarity, open-sourcing of code, and the practical significance of improving throughput.

While the reviewers are generally positive, the consensus strengths are somewhat overshadowed by insufficient baseline comparisons (reviewer `ehFX`), and open concerns about stability (reviewer `rF4g`). These weaknesses are not merely incremental gaps but raise questions about how broadly applicable and reliable the proposed approach will be in practice.

I believe this paper can be further strengthened by broader empirical validation across models and tasks. Based on the current version, I decide to recommend rejection.